# Clinical Outcome of Immediate Transurethral Surgery for Benign Prostate Obstruction Patients with Acute Urinary Retention: More Radical Resection Resulted in Better Voiding Function

**DOI:** 10.3390/jcm8091278

**Published:** 2019-08-22

**Authors:** Liang-Kang Huang, Ying-Hsu Chang, I-Hung Shao, Tsung-Lin Lee, Ming-Li Hsieh

**Affiliations:** 1Division of Urology, Department of Surgery, Linkou Chang Gung Memorial Hospital, Taoyuan City 33302, Taiwan; 2College of Medicine, Chang Gung University, Taoyuan City 33302, Taiwan

**Keywords:** benign prostate obstruction, transurethral resection of prostate, laser vaporization, acute urinary retention, immediate surgery, transurethral surgery

## Abstract

Introduction: Transurethral surgery of the prostate is currently the gold standard treatment modality for patients with benign prostatic hyperplasia (BPH) with recurrent acute urinary retention. This study aimed to evaluate the outcome and predictors of patients receiving immediate surgery after acute urinary retention (AUR) episodes. Materials and Methods: From January 2016 to January 2017, we retrospectively included 714 patients who received transurethral surgery of prostate due to BPH. Among them, 158 patients received surgeries immediately after an AUR episode. General characteristics data including age, Body mass index (BMI), International prostate symptom score (IPSS score), prostate volume and Prostate-specific antigen (PSA) were reviewed. We also collected surgery-related parameters including surgical types, operation time, and specimen weight. Resection ratio was defined as (resected specimen weight)/(Transurethral ultrasound (TRUS) volume). The catheterization status on discharge, post-operative medication for BPH, and AUR within 3 months after operation were evaluated. Statistical analysis was performed with Statistical product and service solutions (SPSS). Results: The mean age of the patients was 73.5 years, with a BMI of 24 kg/m^2^. IPSS total score was 25 with a Voiding symptom/Storage symptom score (V/S score) of 14.6 and 10.4, respectively. A total of 74 (46.8%) patients still took medication for BPH for over 1 month after the surgeries, 28 (17.7%) patients were not catheter-free at the time of discharge, and 14 (8.9%) patients had AUR within 3 months after the surgeries. Surgical type did not impact the outcome of surgeries. In patients who received Transurethral resection of the prostate (TURP), the resection ratio is the only predictor for the catheterization status on discharge and post-operative medication for BPH within 3 months after operation. Conclusions: In patients with BPH-related AUR followed by immediate transurethral surgeries, more radical resection is significantly correlated with short-term medication-free and catheter-free status.

## 1. Introduction

Benign prostatic hyperplasia (BPH) is a pathological condition that has a strong correlation with aging. It causes lower urinary tract symptoms such as urine urgency, frequency, and nocturia that affect an individual’s quality of life. In a previous study, the estimated histological incidence of BPH was approximately 50% in men 50 years of age and 75% in men 80 years of age [1].

In men with BPH, a transitional zone with adenomatous tissue, which surrounds the urethra, grows and compresses it gradually, obstructing the bladder outlet. Therefore, detrusor muscle pressure increases to pass urine smoothly. As BPH progresses, it can result in urinary retention, regardless of whether it is acute or chronic.

Acute urinary retention (AUR) is a urological emergency in which a sudden inability to urinate may be accompanied by severe dysuria, anxiety, and discomfort. Such a critical condition typically requires catheterization through the urethra for urine drainage, treatment with an α-blocker, followed by a trial without catheterization (TWOC), which is the standard treatment worldwide. In a recent study of the Reten-World survey, the overall success rate of TWOC was 61% [2]. After 3 days, the urethral catheters were removed, and over 60% of the patients had recovered the ability to urinate naturally [3]. The TWOC treatment plan is based on previous research studies that suggested that unnecessary urgent surgical intervention in cases of AUR could increase the complication and death rates of patients within 30 days of AUR [4,5]. However, AUR recurrence may be noted by many patients with BPH patients at any time.

Transurethral resection of the prostate (TURP), on the other hand, is still considered the gold standard surgical treatment strategy for BPH and has been for decades. It improves obstructive voiding symptoms and urinary flow rate, with success rates ranging from 85 to 90% [6]. In previous studies of western countries, 20–42% of patients underwent TURP due to AUR [4].

In a previous study by Flanigan et al. [7], patients with BPH were randomized to immediate TURP or watchful waiting. During the study, 36% of patients under watchful waiting underwent TURP within a 5-year follow-up period. In patients who received immediate TURP, peak flow rates, and symptom scores improved more than in those patients who did not undergo TURP. The investigators concluded that delaying operations in patients who needed surgical intervention could cause a worse outcome. Indeed, TURP can cause complications such as postoperative bleeding, incontinence, and the requirement for reoperation. Thus, TURP should be preserved for those with adequate indications, including AUR, renal function deterioration, bladder stone formation, or urinary tract infection. In this study, we aimed to elucidate the role and outcome of immediate surgery in patients who experience acute urinary retention (AUR) episodes.

## 2. Materials and Methods

Patients who presented at the outpatient department or emergency department of a single institution (Chang-Gung Memorial Hospital, Taiwan), from January 2016 to January 2017, were reviewed. The study had been approved by Chang Gung Institutional Review Board (201901289B0), duration of approval is 24 August 2018 to 23 August 2019. We retrospectively reviewed the medical records of 714 patients who underwent TURP or laser prostatectomy. The exclusion criteria were: (1) previous prostate cancer, (2) incomplete or missing data, (3) previous prostate or urethral surgery, (4) diagnosis of neurogenic bladder disorder before AUR episode, and (5) receiving an operation after a period of time of AUR occurrence. Of the 714 patients, 158 patients who underwent surgery immediately after the AUR episode were included in the study. All operations were performed by well-experienced urological attending physicians in Chang-Gung Memorial Hospital. The general characteristics of the patients, including age, body mass index (BMI), International prostate symptom score (IPSS score), prostate volume, T-zone volume, and Prostate-specific antigen (PSA), were collected and reviewed. We also collected surgery-related parameters, including surgical type, operation time, resection ratio, and resected specimen weight. Urodynamic studies such as uroflow or residual urine were not measured because all the patients were under catheter usage status before undergoing surgery.

The resection ratio was defined as (the resected specimen weight)/(the prostate volume measured by Transurethral ultrasound (TRUS)). The resection ratio of laser vaporization with green light was not included in the analysis since no specimen was collected. AUR within 3 months after the operation was evaluated as the primary endpoint, while catheterization status on discharge and alpha-blocker usage 3 months after surgery were set as secondary endpoints, to clarify the immediate recovery condition from surgery. A patient with residual urine over 300 mL detected at the bedside by echography after voiding twice was recommended for discharge with catheterization—either Foley or cystostomy. After operation, treatment with an α-blocker would be continued until Out-patient department (OPD) follow-up one week later after discharge. Patients who were satisfied with voiding the condition at OPD would not be prescribed treatment with an α-blocker. On the other hand, if the patient felt uncomfortable as a result of lower urinary tract symptoms, treatment with an α-blocker would be given based upon the attending physician’s clinical judgement. The statistical analysis was performed using SPSS Ver. 22 (IBM, Chicago, IL, USA).

## 3. Results

From January 2016 to January 2017, there were 158 patients included in this study. Table 1 shows the general characteristics of the patients. Among them, the mean (SD) patient age was 73.46 (8.82) years. Preoperatively, the mean (SD) International prostate symptom score-voiding symptom score (IPSS-V) was 14.61 (3.58), International prostate symptom score-storage symptom score (IPSS-S) was 10.38 (2.53), and the IPSS-T was 25.02 (4.93). The mean (SD) prostate volume estimated on transurethral ultrasound was 63.83 (34.52), and the T-zone volume was 35.87 (21.19). The mean (SD) preoperative PSA was 11.36 (13.23), and the mean (SD) resection ratio was 30.88% (22.73%). In Figure 1, we compared the different types of surgical procedures by the three endpoints including (A) AUR within 3 months after surgery, (B) catheter usage on discharge, and (C) α-blocker usage 3 months after surgery. The surgical types were not associated with AUR or not within 3 months after surgery, catheter usage on discharge, or α-blocker usage 3 months after surgery. There were no statistically significant differences between the groups (all *p* > 0.05).

Among the 158 patients, 14 patients experienced AUR within 3 months after surgery (Table 2). We compared all possible parameters mentioned above for the evaluation of the prediction of AUR occurrence events such as IPSS, prostate volume, PSA, and resection ratio. Although the resection ratio in the AUR group was lower than that in the group without AUR, the difference was not statistically significant.

There were 28 patients assigned to catheter usage on discharge (Table 3). In a comparison of the different predictors for catheter use, the resection ratio was the only factor that influenced assignment to catheterization at discharge or not (*p* = 0.002). The finding for α-blocker usage 3 months after surgery (Table 4) in which 74 patients still took an α-blocker after surgery (*p* = 0.026) was similar, with the resection ratio being less in the group that was still taking α-blockers 3 months after surgery.

A total of 123 patients received regular Alpha-blocker (α-blocker) agents for BPH before the surgeries. The comparisons of outcome between patients with and without previous treatment with an α-blocker in three endpoints are shown in Table 5. Pre-operative α-blocker usage was not associated with AUR occurrence within 3 months, catheter usage on discharge, or post-operative α-blocker usage in our analysis.

Table 6 shows the post-operative complications with urinary tract infection found in 50 patients and hematuria in 52 patients. All patients’ conditions improved after conservative treatment including antibiotics or hydration. Acute urinary retention was noted in 14 patients who were recommended catheterization before discharge, either Foley or cystostomy. Additionally, all the patients were catheter-free within 1 month during OPD follow-up, thus the results reached the main goal of surgeries. A total of 16 patients complained of urge urinary incontinence after operation and symptoms relieved after medication treatment. There were two patients who suffered from urethral stricture and later received optic urethrectomy.

Summarizing the above findings, in patients receiving TURP after an AUR episode, the resection ratio was the only predictor of catheterization status on discharge and α-blocker usage 3 months after surgery.

## 4. Discussion

BPH, which has been identified as the most common disease in older men, progresses very slowly. In America, a 50-year-old man with BPH has a 40% chance of undergoing treatment (regardless of whether it is surgery or medication) in his whole life [8]. AUR is an emergent and highly uncomfortable event, which may occur during the natural progression of BPH. It is often noted in older men, who have higher ASA scores and underlying diseases such as diabetes, which contribute to detrusor hypocontractility. Larger prostate volumes and PSA levels were also noted in men with AUR [9]. However, the definitive pathological reason is obscure and may be composed of many factors. It is thought to be due to a combination of two or more of the following factors including outflow obstruction (BPH, blood clot obstruction, and urethral stricture), overdistention of the bladder (constipation, decreased bladder contractility, high alcohol, tea or coffee intake, and immobility of the bladder), neuropathic impairment (multiple sclerosis and diabetic cystopathy), or dynamic obstruction (prostate inflammation and α-adrenergic activity enhancement) [10,11,12]. Of these factors, outflow obstruction is the most common [13].

Once AUR occurs, patients may need urgent catheterization to relieve pain and empty the bladder, regardless of an indwelling urethral Foley or suprapubic cystostomy catheter. Subsequent treatment including medical treatment or surgical management might need to be considered, although patients with UR induced by BPH or bladder outlet obstruction (BPO) make up most of the patients for whom TURP is indicated [14]. Nonetheless, it not the first treatment choice in current daily practice because such surgery may cause potential risks and complications [15]. On the other hand, initial catheterization followed by medical treatment with α-blockers is considered the first-line management for BPH and BPO [15]. Through attenuating the sympathetic tone of the urethra and bladder neck, α-blockers reduce bladder outlet resistance to help the patient return to normal voiding [2].

There is no consensus regarding further treatment plans, including when the catheter should be removed after initial management, when patients should undergo TURP or medical treatment only, when surgery be performed, whether α-blockers should be given or not, and for how long, and when to consider surgery again if conservative treatment fails [2,16,17,18]. The Reten-World survey has shown that the success rate of TWOC was better in men who were treated with an α-blocker before removal of the catheter than those who were not so treated, regardless of the duration of catheter insertion. Also, a further analysis confirmed that a few factors influence the success rate or TWOC such as age younger than 70 years, mild lower urinary tract symptoms, a prostate size smaller than 50 g, less than 1000 mL urine drained during catheterization, and provoked AUR compared with spontaneous AUR. On the other hand, although the longer TWOC times coordinate with α-blocker usage can increase the success rate of TWOC, the long duration of TWOC also increases the risk of complications. There are significantly more complications, including hematuria, bacteriuria, urinary tract infection, urosepsis, catheter obstruction, and urine leakage in patients with catheterization lasting over 3 days [2].

Although some troubles related to catheters could be eliminated by successful TWOC, there are still unknown questions such as could patients void smoothly by themselves? How much quality of life they have they gained? How many symptoms were reduced? Would AUR recur in the future [19]? TURP is still the gold standard treatment for those who have failed self-voiding despite medication or temporary catheterization. Ankur et al. showed that age older than 65 years, initial IPSS score >20, intravesical prostatic protrusion >9 mm, prostate volume >56 mL, or residual urine after voiding >750 mL could have a lower chance of successful TWOC after AUR occurs in patients with BPH. Thus, immediate surgery following AUR may be considered [20].

A recent systemic review conducted by Karavitakis et al. focused on the management of urinary retention or benign prostate obstruction suggested that α1-blockers (such as Alfuzosin and Tamsulosin) may improve the outcome of UR or BPO. However, most nonpharmacological treatments, including surgeries, had not been well evaluated in patients with BPO-related urinary retention [21].

In our study, all patients underwent TURP or laser prostatectomy immediately after the occurrence of an AUR episode. We aimed to discern the predictors for the three endpoints, including catheter usage at discharge, AUR recurrence, and BPH drug usage 3 months after surgery. Age, BMI, IPSS, prostate volume, PSA, and prostate resection weight did not interfere with the three endpoints. However, the resection ratio of the prostate influenced catheter and α-blocker usage. The higher the resection ratio was, the lower the chance the patient might need catheterization at discharge or further medical treatment. The higher resection ratios were related to higher changes in the immediate postoperative catheter-free and short-term medication-free conditions. However, relative long-term follow-up for recurrent AUR was not associated with the resection ratio. Our data also corresponded to previous studies showing that men who had an AUR episode were older, had a larger prostate volume, and higher PSA levels than those who did not have an AUR event [22].

In this study, we highlight some limitations. First, this is a retrospective study with relatively small group sizes. Second, preoperative urodynamic studies were not obtained because all the patients were catheterized and underwent surgical intervention immediately after the occurrence of AUR. Third, some other parameters, such as smoking, drinking, exercising, and comorbidities, were not collected. Fourth, we analyzed the patient’s condition within 3 months of surgery; nonetheless, it would have been more thorough if we had documented a longer follow-up. However, to our knowledge, this is the first study to investigate the notion that the resection ratio may influence the outcome of patients undergoing surgical intervention. Higher resection ratios may lower catheterization rates on discharge and short-term α-blocker usage after surgery.

## 5. Conclusions

TURP or laser prostatectomy provides favorable long-term outcomes in patients with AUR. Routine urodynamic or uroflow studies are not necessary for adequately selected patients. More radical resection of the prostate tissue during surgery reduces the need for catheterization on discharge and α-blocker usage after surgery. In the future, better controlled, randomized, prospective trials in larger patient populations and with longer-term follow-up are needed to confirm our study results.

## Figures and Tables

**Figure 1 jcm-08-01278-f001:**
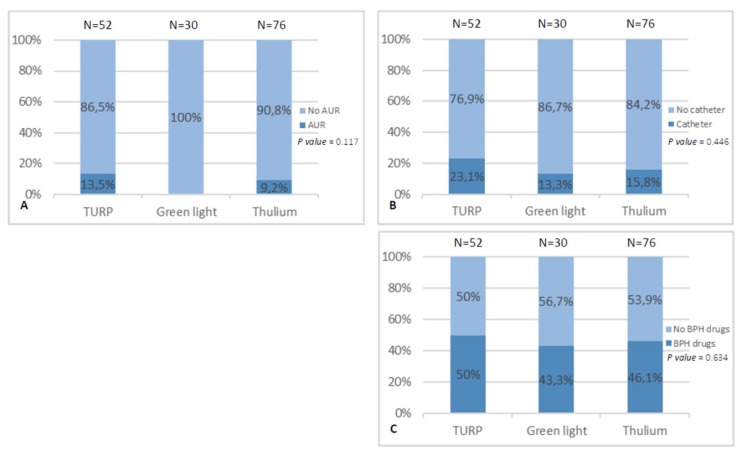
Surgical types versus three end points: (**A**) acute urinary retention (AUR) within 3 months after the operation; (**B**) catheterization status on discharge; (**C**) α-blocker usage 3 months after surgery.

**Table 1 jcm-08-01278-t001:** Basic patient characteristics.

Variable	Mean (SD)
No. of patients (%)	158 (100%)
Age (years)	73.46 (8.82) (52–93)
BMI (kg/m^2^)	23.99 (3.55) (15.82–34.67)
IPSS_V	14.61 (3.58) (2–20)
IPSS_S	10.38 (2.53) (3–16)
IPSS_T	25.02 (4.93) (9–35)
TRUS_TPV (mL)	67.41 (28.66) (15.92–178)
TRUS_T zone (mL)	35.87 (21.19) (0–119.6)
PSA (ng/mL)	11.36 (13.23) (0.41–91.21)
Operation time (minutes)	129.91 (47.32) (36–283)
Resection ratio (%)	30.88 (22.73) (1–95)

BMI: body mass index, IPSS_S: International prostate symptom score (IPSS) storage symptom scores, IPSS_T: IPSS total scores, IPSS_V: IPSS voiding symptom scores, PSA: prostate-specific antigen, TRUS_TPV: TRUS estimated total prostate volume.

**Table 2 jcm-08-01278-t002:** AUR within 3 months after surgery.

Variable	AUR		*p* Value
	Yes	No	
No. of patients (%)	14 (8.9)	144 (91.1)	
Age (years)	76.29 (11)	73.18 (8.58)	0.21
BMI (kg/m^2^)	23.63 (3.39)	24.02 (3.57)	0.7
IPSS_V	13.6 (2.41)	14.71 (3.67)	0.35
IPSS_S	10.7 (1.77)	10.35 (2.59)	0.68
IPSS_T	24.3 (3.86)	25.08 (5.03)	0.63
TRUS_TPV (mL)	63.83 (34.52)	67.76 (28.16)	0.64
TRUS_T zone (mL)	32.14 (25.96)	36.25 (20.75)	0.54
PSA (ng/mL)	6.11 (5.46)	11.85 (13.64)	0.15
Operation time (minutes)	115.93 (56.81)	131.28 (46.3)	0.25
Resection ratio (%)	28.15 (21.43)	34.84 (23.24)	0.33
Resection weight (g)	15.07 (13.15)	24.43 (21.46)	0.31
AUR (no. episodes)	1.64 (0.84)	1.5 (0.84)	0.55

AUR: acute urinary retention, BMI: body mass index, IPSS_S: IPSS storage symptom scores, IPSS_T: IPSS total scores, IPSS_V: IPSS voiding symptom scores, PSA: prostate-specific antigen, TRUS_TPV: TRUS estimated total prostate volume.

**Table 3 jcm-08-01278-t003:** Catheter usage on discharge.

Variable	Catheter		*p* Value
	Yes	No	
No. of patients (%)	28 (17.7)	130 (82.3)	
Age (years)	76.29 (9.77)	72.85 (8.52)	0.06
BMI (kg/m^2^)	24.06 (3.79)	23.97 (3.51)	0.91
IPSS_V	15 (2.99)	14.53 (3.71)	0.58
IPSS_S	10.32 (2.32)	10.39 (2.59)	0.9
IPSS_T	25.32 (4.17)	24.95 (5.11)	0.85
TRUS_TPV (mL)	75.31 (36.2)	65.71 (26.64)	0.12
TRUS_T zone (mL)	40.94 (28.53)	34.53 (18.73)	0.18
PSA (ng/mL)	11.58 (10.63)	11.32 (13.77)	0.93
Operation time (minutes)	142.71 (49.4)	127.13 (46.59)	0.12
Resection ratio (%)	24.26 (13.1)	36.48 (24.33)	0.002
Resection weight (g)	19.88 (16.89)	24.32 (19.88)	0.34
AUR (no. episodes)	1.61 (0.92)	1.49 (0.83)	0.52

AUR: acute urinary retention, BMI: body mass index, IPSS_S: IPSS storage symptom scores, IPSS_T: IPSS total scores, IPSS_V: IPSS voiding symptom scores, PSA: prostate-specific antigen, TRUS_TPV: TRUS estimated total prostate volume.

**Table 4 jcm-08-01278-t004:** α-blocker usage 3 months after surgery.

Variable	α-Blocker		*p* Value
	Yes	No	
No. of patients (%)	74 (46.8)	84 (53.2)	
Age (years)	74.69 (9)	72.4 (8.62)	0.105
BMI (kg/m^2^)	24.09 (3.56)	23.92 (3.57)	0.764
IPSS_V	15.12 (2.85)	14.02 (4.1)	0.093
IPSS_S	10.63 (2.4)	10.1 (2.66)	0.263
IPSS_T	25.75 (3.91)	24.18 (5.66)	0.082
TRUS_TPV (mL)	69.46 (30.49)	65.8 (27.26)	0.446
TRUS_T zone (mL)	37.89 (23.66)	34.27 (18.94)	0.358
PSA (ng/mL)	11.79 (13.45)	11.06 (13.2)	0.746
Operation time (minutes)	129.86 (52.4)	129.96 (43)	0.101
Resection ratio (%)	29.04 (17.92)	38.27 (26.2)	0.026
Resection weight (g)	20.26 (15.87)	26.34 (24.45)	0.095
AUR (no. episodes)	1.44 (0.76)	1.59 (0.92)	0.283

AUR: acute urinary retention, BMI: body mass index, IPSS_S: IPSS storage symptom scores, IPSS_T: IPSS total scores, IPSS_V: IPSS voiding symptom scores, PSA: prostate-specific antigen, TRUS_TPV: TRUS estimated total prostate volume.

**Table 5 jcm-08-01278-t005:** Pre-operative (Pre-op) α-blocker usage with end points.

Pre-op α-Blocker Usage	Yes		No		*p* Value
	123		35		
	Yes	No	Yes	No	
AUR within 3 months	10 (8.13)	113 (91.87)	4 (11.43)	31 (88.57)	0.376
Post-operative catheter usage	20 (16.26)	103 (83.74)	8 (22.86)	27 (77.14)	0.252
Post-operative BPH drugs usage	59 (47.97)	64 (52.03)	15 (42.86)	20 (57.14)	0.367

**Table 6 jcm-08-01278-t006:** Post-operative (post-op) complications.

Post-op Complications	Numbers (Percentage)	The Clavien Dindo Classification Grade
Urinary tract infection	50 (31.65)	II
Hematuria	52 (32.91)	I
Acute urinary retention	14 (8.86)	III
Urge urinary incontinence	16 (10.13)	II
Urethral stricture	2 (1.27)	III

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
