# Peer review of "Clinical Outcome of Immediate Transurethral Surgery for Benign Prostate Obstruction Patients with Acute Urinary Retention: More Radical Resection Resulted in Better Voiding Function"

_jcm, 2019, doi:10.3390/jcm8091278_

Round 1

Reviewer 1 Report

Dear Editor,

thank you for possibility to review manuscript “Clinical outcomes of immediate transurethral surgery for benign obstruction patients with acute urinary retention: more radical resection brought better voiding function”

The main goal for BPH surgery is to remove as much as possible obstructed tissue and treatment effectiveness directly depends on resected tissue weight ratio with total or even better with transition zone volume. So authors assertion that there are no data on that is not correct (line197-199) and this topic is not new.

Criteria of effectiveness for BPH surgery has been suggested 2 decades ago by Second International Consultation on Benign Prostatic Hyperplasia - pre and postoperative difference of IPSS, QoL, Qmax and prostate volume. Authors selected end-points for study are controversial:

1.” Did not void well” and “too much residual volume” (line 81-82) are insufficient for strict decision to discharge patient with catheter.  

2. “usage alpha blockers at 3 months after surgery” (line 79) also very inaccurate without strict explanation when medication should be stopped or start to reuse.

3. only AUR episode during 3 months after surgery should be used as end-point.

Statistical analysis for proper conclusions is insufficient – logistic regression analysis in such study is necessary.  

Question to do or not immediate surgery after AUR is interesting but most clinicians are worried in possible higher complications rate and less efficacy comparing with planned surgery. Radicality of surgery, in my opinion, is recognized criteria for efficacy.

Author Response

Thanks for the reviewer doctor’s patience of reading whole article and raise these helpful advises for our article. We have corrected our article according to the reviewer's advise. And we would have step by step answers and improvement that showing as below:

Point 1: The main goal for BPH surgery is to remove as much as possible obstructed tissue and treatment effectiveness directly depends on resected tissue weight ratio with total or even better with transition zone volume. So authors assertion that there are no data on that is not correct (line197-199) and this topic is not new.

A: The critical point of this study was the emphasis of immediate transurethral surgery after AUR episode because of benign prostate obstruction. Different from BPH, previous articles did not mention this goal. Since AUR correlates with multiple factors more than just BPH that we have mentioned in the article. We wanted to find out the parameters that might influence the prognosis of operations after AUR occurred.

Point 2: Criteria of effectiveness for BPH surgery has been suggested 2 decades ago by Second International Consultation on Benign Prostatic Hyperplasia - pre and postoperative difference of IPSS, QoL, Qmax and prostate volume. Authors selected end-points for study are controversial

A: Indeed, the criteria of effectiveness for BPH surgeries were discussed for a long period of time and some precious findings were announced in previous articles. As for our study, all of the 158 patients were able to void by themselves after operations during OPD follow-up. Though some patients might suffered from episodes of UTI or hematuria related with AUR. They could self void at a later time after conservative treatment. So that means all the patients reached primary goal of surgeries. Thus we set the primary end point as AUR recurrence within 3 months. And 2 other secondary points as catheter status on discharge and alpha-blocker usage within 3 months. And we wanted to find out which parameters would influence primary and secondary endpoints we supposed in the study.

Point 3: ” Did not void well” and “too much residual volume” (line 81-82) are insufficient for strict decision to discharge patient with catheter.  

A: Thanks for the review doctor’s helpful advise, we have corrected the obscure descriptions in article with definition that showed below: patients who had residual urine more than 300 ml detected by bedside echography after voiding twice were recommended of catheterization, either Foley or cystostomy.

Point 4: “usage alpha blockers at 3 months after surgery” (line 79) also very inaccurate without strict explanation when medication should be stopped or start to reuse.

A: After operation, α-Blocker would be continued until OPD follow-up a week later after discharge. Patients who were satisfied with voiding condition at OPD would not be prescribed with α-Blocker. On the other hand, if the patient felt bothersome with lower urinary tract symptoms, α-Blocker would be given based upon attending physician’ s clinical judgement.

Point 5: only AUR episode during 3 months after surgery should be used as end-point.

A: Thanks for reviewer doctor’s professional opinion. As we had mentioned at previous period, since all patients achieved primary goals of TUR surgeries. We tried to find out more parameters that might influence other clinical outcomes of operations. Thus we set AUR within 3 months as primary endpoint while catheter status on discharge and α-Blocker usage within 3 months after surgeries as secondary endpoints.

Point 6: Statistical analysis for proper conclusions is insufficient – logistic regression analysis in such study is necessary.  

A: We used single-variant analysis for parameters that correlates with endpoints we set. Resection ratio was the only factor that might associated with status on discharge and α-Blocker usage within 3 months after surgeries. Thus to our disappointment, multi-variant analysis was not suitable for our study.

Reviewer 2 Report

The authors try to investigate Clinical Outcome of Immediate Transurethral 2 Surgery for Benign Prostate Obstruction Patients with 3 Acute Urinary Retention: More Radical Resection 4 Brought Better Voiding Function 

 The purpose of the investigation is justified. However, Major deficiencies in methodology that  implicated on the study. 

1-     The number of patients is too small  to reach a significant conclusion 

The authors started with 714 pt for a study period  that is more than five years .Although they have a subgroup of 158 patients in 12 months !

2-      Study design is retrospective study  and It is confusing  about the number of the patients. I think  only the group of 158  patients should be included in the study. The should only include TURP group.

3 - The authors failed to identify the success criteria after the surgery . They mentioned the success rate was if the pt do not need alpha blocker or does not have AUR in 3 months . You can feel that the authors did not pay enough attention to this point which is a critical point in  this study . 

4- How many surgeons were involved in the study and what was their level of expertise 

5- They reported in page 2  ( urodynamic study was not performed in all pt . they should explain what their protocol regarding UDS prior to TURP.  

6-       The authors did not use Clavien Dindo classification to assess  post procedure  complications.

7- In page 2 they report that  ( The patient 81 who did not void well or who had too much residual urine detected at the bedside by echography 82 after voiding was recommended for discharge with catheterization,.) without justify if these pt consider their outcome 

Author Response

Thanks for the reviewer doctor’s patience of reading whole article and raise these helpful advises for our article. We have corrected our article according to the reviewer's advise. And we would have step by step answers and improvement that showing as below:

Point 1: The number of patients is too small to reach a significant conclusion. The authors started with 714 pt for a study period that is more than five years. Although they have a subgroup of 158 patients in 12 months!

A: The database of this article was from patients of CGMH hospital, from Jan 2016 to Jan 2017. There were total 714 patients received TURP or laser prostatectomy during 1 year period. Among them, 158 were AUR patients who received operation immediately. Thus there were only 158 patients enrolled into this study with further analysis.

Point 2: Study design is retrospective study and It is confusing about the number of the patients. I think only the group of 158 patients should be included in the study. The should only include TURP group.

A: Thanks again for professional suggestions. There were 158 patients who received TURP or laser prostatectomy immediately after AUR episode, which constituted our database.

Point 3: The authors failed to identify the success criteria after the surgery. They mentioned the success rate was if the pt do not need alpha blocker or does not have AUR in 3 months. You can feel that the authors did not pay enough attention to this point which is a critical point in  this study. 

A: Thanks for the precious opinions of our article, in our study, all of the 158 patients were able to void by themselves after operations during OPD follow-up. Though some patients might suffered from episodes of UTI or hematuria related with AUR. They could self void at a later time after conservative treatment. So that means all the patients reached primary goal of surgeries. Thus we set the primary end point as AUR recurrence within 3 months, while 2 other secondary points as catheter status on discharge and alpha-blocker usage within 3 months. And we wanted to find out which parameters would influence primary and secondary endpoints we supposed in the study.

Point 4: How many surgeons were involved in the study and what was their level of expertise.

A: All operations were performed by 21 well-experienced urological attending physicians who have performed at least 50 successful TURP or laser prostatectomy.

Point 5: They reported in page 2  ( urodynamic study was not performed in all pt ). hey should explain what their protocol regarding UDS prior to TURP.  

A: Because all patients were under catheter status (no matter Foley or cystostomy), UDS was not performed considering patients suffering from repeated catheter insertion.

Point 6: The authors did not use Clavien Dindo classification to assess post procedure  complications.

A: Thanks for reminding of the neglects. The complications with Clavien Dindo classification was set as table 5 for correction.

Point 7: In page 2 they report that  ( The patient 81 who did not void well or who had too much residual urine detected at the bedside by echography 82 after voiding was recommended for discharge with catheterization ) without justify if these pt consider their outcome.

A: That was our mistake that we did not mention clearly at previous article. After removal of Foley, all patients would try voiding and received bedside echography. Those who have residual urine more than 300 ml after voiding twice were recommended of catheter on discharge.

Thanks for kindly reading of answers above.

Round 2

Reviewer 1 Report

Methodologically presented study constructed not correctly. The main goals of surgery are:

1.     to improve Qmax,

2.     to decrease symptoms (IPSS),

3.     to improve the QoL,

4.     to reduce post voiding residual urinary volume.

These parameters are usually investigated in BPH studies. Presented  in manuscript results could be hardly comparable to other published data. The results should be investigated at least 6 month after initiated treatment and 1-year period is optimal for short term effectiveness. Catheter-free after surgery end point is not conclusive, because heeling of tissue after surgery takes up to one month and depends on a lot of factors. It only could be as predictor for results at  6 or 12 months.

Medication-free within end point at 3 post-operative months is very subjective and without very strict schedule of visits cannot say anything about effectiveness of surgery.

Only AUR could be used as end point, but in literature it presented very rear and usually not as the main end point, so comparison presented data to other also will be complicated.

Statistic of presented data could be much larger (for example - logistic regression).

Author Response

First of all, We appreciated your valuable comments for our article, the responses to your comments were listed below:

As the parameters which the reviewer suggested such as Qmax, IPSS, Or QoL. Because most of the patients who suffered from AUR initially presented at ER longed for immediate treatment for critical problems. All of them received suprapubic cystostomy or Foley insertion as urgent procedures. The timing was not suitable for questionnaires such as IPSS or QoL. Qmax was not examined either since all of them were under catheter status after emergent treatment. 

Indeed, there were many published studies which have thoroughly researched about risk factors causing lower urinary tract symptoms, surgical types comparisons, parameters associated with outcomes, or complications of surgeries for BPH, no matter TURP, laser prostatectomy using enucleation or vaporization. BPH or BPO is a multi-factorial disease which many causes interacting. And it takes half to 1 year follow-up to see the outcomes after operations. To our acknowledgement, there was no paper solely studied about parameters for outcomes of AUR patients who received immediate surgeries. And we only took short-term outcomes in a 3-month period because patients who were catheter-free or medication-free might not do regular follow-up since such patients thought they were not bothered with this disease anymore.

Some of the patients were still under regular follow up at our hospital. And we would launch the next research with the emphasis on these patients for further investigations.

And thanks for the opinions about statistical methods improvements. To our disappointment, the resection ratio was the only parameter left after single-variant analysis of our study. So multi-variant analysis or logistic regression might not be suitable with our topic.

If you have any further suggestions for changes, please let us know. Great thanks for your kindly advises.

Reviewer 2 Report

I will clearly separate  TURP group from laser group 

Author Response

First of all, We appreciated your valuable comments for our article, the responses to your comments were listed below:

TURP was more similar with laser prostatectomy using enucleation rather than laser vaporization. So we eliminated laser vaporization at initial since it was a different surgical technique and there would be no specimen left for further investigations. And because there were many published papers researching about safety, complications, prognosis of laser prostatectomy with comparisons of TURP. In this article, we might not divide laser prostatectomy from TURP, but see them as the same group instead.

If you have any further suggestions for changes, please let us know. Great thanks for your kindly advises.